# "Crises are a perpetual restart"—A comparative analysis of maternal and newborn health political prioritization across four fragile and conflict affected settings

**Mamothena Carol Mothupi** [1]*, **Teresia Macharia**[1], **Katja Starc Card**[2], **Rosine Bigirinama**[3], **Alicia Adler**[4], **Maryan Abdulkadir Ahmed**[5], **Abdirisak Dalmar** [5], **Rifkatu Aimu Sunday**[6], **Sussan Israel-Isah** [6], **Kon Paul Alier** [7], **Naoko Kozuki**[8], **Paul Spiegel**[9], on behalf of the EQUAL Consortium¶

**1** International Rescue Committee, Nairobi, Kenya, **2** International Rescue Committee, Geneva, Switzerland, **3** Ecole Régionale de Santé Publique, Université Catholique de Bukavu, Bukavu, Democratic Republic of Congo, **4** International Rescue Committee, New York, New York, United States of America, **5** Somali Research and Development Institute (SORDI), Mogadishu, Somalia, **6** Institute of Human Virology Nigeria, Abuja, Nigeria, **7** Department of Community Medicine, School of Medicine, University of Juba, South Sudan, **8** International Rescue Committee, New York, New York, United States of America, **9** Johns Hopkins Bloomberg School of Public Health, Johns Hopkins University, Baltimore, Maryland, United States of America

¶ Membership of the EQUAL Consortium Working Group is listed in the Acknowledgments.
* mamothena.mothupi@rescue.org

## Abstract

Fragile settings account for the highest burden of maternal and neonatal mortality in the world. Although policies and strategies for intervention exist, there are gaps on political prioritization of maternal and newborn health (MNH) by key actors to maximize resources and impact of programs. This study offers a comparative analysis of MNH political prioritization across four fragile national and sub-national settings, and recommendations for improving policies and strategies. Primary data from descriptive case studies conducted in Somalia, Nigeria, Democratic Republic of Congo and South Sudan was analysed comparatively using the Shiffman and Smith (2007) political prioritization framework. In total, eighty-two (82) interviews were conducted with government, international multilateral and non-governmental organizations, national civil society organization, public and private health providers, funders and academic key informants. Across settings, MNH was seen as priority despite the crises and the need to strengthen the health system as a whole. Crisis conditions also created policy windows for transformation of health systems, as evidenced in investments towards universal health coverage (free maternal health services) and community health systems. Transformations were enabled by political championship by local actors, aligned with technical planning, national priorities, and international momentum. Conversely, external actor and budget politics had a negative effect on regulation of the private health sector, inclusivity in decision-making, resource allocation,

**Data availability statement:** The interviews conducted across the four countries were with key Ministry of Health, Ministry of Finance, donors, and other high-level political stakeholders. The issue involves political prioritization of a health program, and touches on political will (or lack thereof), government influence (sometimes negative), and power dynamics between health actors. We are concerned about the ability to adequately anonymize the interview transcripts for public sharing in a repository. Some of the countries we did the analysis in hold us to extremely high standards for confidentiality. We are, however, willing to share data on a case-by-case basis, assessing each request to ensure proper anonymization of data before sharing and in accordance with the request. Our institutional contact for sustainable data access is the International Rescue Committee IRB chair and administrator, accessible through this email (humansubjects@rescue.org), and the current chairperson is Dr Elizabeth Radin (elizabeth.radin@rescue.org,) who is also the Senior Director for Research and not a member of this research team or co-author.

**Funding:** This research was funded by UK International Development from the UK government as part of the EQUAL Research Programme Consortium (PO 8613 to NK). The funders had no role in study design, data collection and analysis, decision to publish, or preparation of the manuscript.

**Competing interests:** The authors have declared that no competing interests exist.

and policy community cohesion. International actors held significant influence in producing evidence for MNH, mobilizing and distributing resources, and technical program planning.Concerns were raised about their negative impact on local capacities, sustainability, and geographical distribution of services. Across settings, key informants emphasized how involvement of local organizations, civil society, religious and community leaders could enhance accountability, demand for care and political prioritization of MNH. Addressing gender inequities at the provider-patient interface, health facility management and high-level decision-making could also enhance political prioritization and contextualization of programs.

## Introduction

Fragile countries and regions face disproportionately high burdens of maternal and newborn morbidity and mortality. These settings made up the majority of the top 15 countries with the highest neonatal mortality rates in 2018 [1]. According to the World Health Organization (WHO), the average maternal mortality ratio in fragile countries is 551 deaths per 100,000 live births, which was double the world average in 2020 [2]. Fragile countries in Africa have some of the worst indicators, with South Sudan, Chad and Nigeria having maternal mortality ratios above 1,000 deaths per 100,000 live births [2]. The African continent also has the highest under five mortality rates in the world, a significant portion of which is constituted by newborn deaths [3]. The annual rate of reduction in neonatal mortality was also one of the slowest globally, in the decades since 1990 [3].

Fragility is defined as "the combination of exposure to risk and insufficient coping capacities of the state, system and/or communities to manage, absorb or mitigate those risks" [4]. These risks and capacities are considered across multiple dimensions encompassing economic, environmental, security, societal, human and political factors. About a quarter of the world's population and the majority of the world's extremely poor live in the 60 fragile states [3]. Due to fragility, health indicators for the population are poorer and access to care is negatively affected. According to the Organisation for Economic Co-operation and Development, no fragile context is on track to achieve health-related Sustainable Development Goals [4]. This is particularly true for women and children's health targets. A WHO report shows how complex crises such as conflict, the COVID-19 pandemic and climate change led to dramatic increases in maternal mortality due to disruptions in comprehensive interventions [5]. Fragility also affects the ability of the state to develop policies and manage public resources [6]. While some fragile contexts may have reproductive, maternal, newborn and child health policies in place, implementation is often weak due to insufficient capacities [7]. Governments in fragile states often play a stewardship role with non-governmental actors implementing policies [8]. Nonetheless, political commitment has been identified as a necessary driver of rebuilding health systems in fragile states [8]. Political commitment is essential to political prioritization of the health

agenda, to restore implementation and resource allocation to government where feasible and enhance accountability for health outcomes.

Health systems in fragile contexts are characterized by several challenges including health worker displacement, fragmented service delivery, limited access to care, poor regulation of health providers, weakened governance and lack of financial resources [9,10]. Fragile countries in Africa have the lowest midwife density globally [11]. Health system resilience in these contexts can be assured by strong capacities for governance especially during conflict, and better integration of community-based interventions [12]. Good governance is associated with political will and better prioritization of health responses, aligned to populations needs and preferences. This alignment of priorities is key to better legitimacy of health institutions and management of actors that provide health services in fragile settings [12], both of which have a bearing on the coverage and uptake of health services. Thus enhanced political will and prioritization of maternal and newborn health could lead to better access to services and improvement of outcomes in these settings.

There are still evidence gaps in MNH in fragile settings regarding policies, strategies and effective interventions that can improve outcomes. A systematic review of conflict affected settings found postnatal care, newborn care and community engagement as some of the key gaps in evidence [13]. The breadth of settings of available evidence is also limited, as the majority studied displaced populations in camps and not national health systems. In addition, the thinking and approaches adopted by national level policy makers, implementers, advocates and other stakeholders is not adequately reflected in the current evidence [14]. National stakeholders can play a key role in domestication of international commitments and expanded coverage and sustainability of interventions, given adequate political will, leadership and resources. It is thus imperative to understand factors driving political prioritization of MNH among key stakeholders in fragile countries and provide recommendations for improved attention and action to address the high burden of morbidity and mortality across similar contexts. This study compares national and sub-national stakeholder perspectives in political prioritization of MNH across four fragile settings in Africa and proposes recommendations for improving policy and action across key elements.

## Methodology

### Ethics statement

The study was approved by the International Rescue Committee Institutional Review Board (approval number H.1.00.052). This study was a secondary data analysis and not primary data collection therefore consent was not sought directly from the participants of the original studies.*Study Design*

We conducted a comparative analysis of primary data from four descriptive case studies of fragile and conflict affected settings in South Sudan (national) [15], Somalia (national), Democratic Republic of Congo (national and the Kivu provinces) [16], and northern Nigeria (Yobe State) [17]. The original case studies explored MNH prioritization broadly in each setting, using the comprehensive health policy analysis triangle [18]. In this secondary analysis, the Shiffman and Smith (2007) framework [19] was used to isolate key components related to *political prioritization* and analyse data comparatively. The four settings were compared on key framework components, which are *ideas, actor power*, *political context* and *issue characteristics* shaping political prioritization. The Shiffman and Smith (2007) framework is an effective tool to use for cross-country comparisons of health political prioritization [20].

### Conceptual framework

The Shiffman and Smith (2007) framework outlines determinants of political concern, policy action and resource allocation for health agendas in global and national contexts. Table 1 describes how the Shiffman framework was adapted to our study. Data from the original studies enabled analysis of all components of the framework, except for external framing due to insufficient data.

**Table 1. Application of the Shiffman and Smith (2007) framework to the comparative analysis.**

| Category[*] | Description[*] | Factors shaping political priority [*] | Adaptation to the comparative analysis |
|---|---|---|---|
| *Actor power* | The strength of the individuals and organizations concerned with the issue | *Policy community cohesion*: the degree of coalescence among the network of individuals centrally involved with the issue at the global level | Degree of coalescence among the network individuals and organizations centrally involved with the issue at national and state level |
| | | *Leadership*: the presence of individuals capable of uniting the policy community and acknowledged as particularly strong champions for the cause | The presence of individuals/stakeholder groups capable of uniting the policy community and acknowledged as particularly strong champions for the cause |
| | | *Guiding institutions:* The effectiveness of organizations or coordinating mechanisms with a mandate to lead the initiative | *Guiding institutions:* The effectiveness of organizations or coordinating mechanisms with a mandate to lead the initiative |
| | | *Civil society mobilization:* The extent to which grassroots organizations have mobilized to press international and national political authorities to address the issue at the global level | Civil society role in policy and action on MNH and linkages and collaboration with other stakeholders |
| *Ideas* | The ways in which actors understand and portray the issue | *Internal frame:* The degree to which the policy community agrees on the definition of, causes of and solutions to the problem. | The ways in which stakeholders understand and portray MNH as a priority |
| | | *External frame:* Public portrayals of the issue in ways that resonate with external audiences, especially the political leaders who control resources | Not applied |
| *Political contexts* | The environments in which actors operate | *Policy windows*: Political moments when global conditions align favorably for an issue, presenting opportunities for advocates to influence decision-makers | Political moments when national conditions (and technical resources) align favorably for an issue, presenting opportunities for advocates and other decision makers to influence policymakers and act on an issue |
| | | *Global governance structure:* The degree to which norms and institutions operating in a sector provide a platform for effective collective action | *National and state level governance structures*: The degree to which norms and institutions operating in a sector provide a platform for effective collective action |
| *Issue characteristics* | Features of the problem | *Credible indicators:* Clear measures that demonstrate the severity of the problem and that can be used to monitor progress | Clear measures that demonstrate severity and monitor progress, as well as the role of research and evidence in decision making within and across organizations. |
| | | *Severity:* The size of the burden relative to other problems, as indicated by objective measures such as mortality levels | The severity of the burden and stakeholder perspectives about MNH as a priority and the impact of crisis on political prioritization. |
| | | *Effective interventions:* The extent to which proposed means of addressing the problem are clearly explained, cost-effective, backed by scientific evidence, simple to implement, and inexpensive | Stakeholder perspectives of conditions for effectiveness, provider capacity, quality of care (including feedback mechanisms) and how their implementation may address maternal and newborn morbidity and mortality |

[*]Shiffman and Smith 2007

## Study settings

Multiple factors were considered as a basis for comparison across the four settings, including the political context for fragility, burden of maternal and neonatal mortality, and socioeconomic indicators. These factors are summarized in S1 Table. Three of the four countries are ranked among the top four most fragile globally in 2023, due to conflict and insecurity [4]. The fourth, Nigeria, is ranked 15th globally. Issues such as ethnic and religious tensions, political disputes, resource competition, human rights violations, and humanitarian crises contribute to the fragility observed in these countries [21,22].

Maternal mortality ratios are very high in all four countries, ranging from 547 deaths per 100,000 live births in the DRC to 1223 deaths per 100,000 live births in South Sudan: similarly, neonatal mortality rates range from 26 deaths per 1,000 live births in the DRC to 40 deaths per 1,000 live births in South Sudan [2,23–25]. Moreover, these statistics may be

underestimated, as low- and middle-income countries often face difficulties in their health surveillance systems, including under-reporting and inconsistent data collection [26]. Using the typology of health systems under stress, as proposed by Pavignani & Colombo (2016) [27], we observe politically legitimate but weakly capacitated governments, some of which are under contestation in regions (Somalia) [28].

Although the gross domestic product (GDP) of the four countries varies greatly, they all share a common concern of high percentages of their population living in poverty. All four countries are classified as multidimensionally poor, facing challenges in health, education, and living standards [29]. Thus, all four countries have low human development index levels, ranging from 0.361 in Somalia to 0.54 in Nigeria, which is below the global average of 0.737 in 2019. These four countries rank among the bottom 20 in the Human Development Index [30–32].

## Data collection

This study was a comparative analysis using transcripts from four original studies. In the original studies, primary data collection was conducted in the four countries at national level for Somalia and South Sudan, national and provincial level for DRC and Yobe State for Nigeria. In each country, semi-structured key informant interviews were conducted with stakeholders including health providers, donors and funders, non-government organizations (NGOs), civil society, Ministries of Health, health facilities and other government ministries. The interviews covered contextual, policy process and content, and actor related factors driving political prioritization of MNH policies and action. Data collection occurred between October to December 2022 in Somalia, November 2022 to January 2023 in Nigeria, October 2022 to August 2023 in DRC, and February to October 2023 in South Sudan. Interviews were conducted by trained qualitative researchers in English and local languages as preferred by respondents. The majority of interviews were recorded (with exception of 2 from South Sudan, where detailed notes were taken). Interviews were transcribed in the language of the interview, translated to English, and uploaded into Dedoose software for analysis. All interviews from these initial studies, a total of 82 interviews was included in this comparative analysis. Table 2 details the numbers and types of participants in each country. For this current comparative analysis, de-identified transcripts were sent to the first author from 14th August 2023 for analysis. Analysis was conducted from May 2024 to December 2024.

## Data analysis

Our study adopted a deductive framework analysis approach. The Shiffman and Smith framework was used to create the initial overarching themes and sub-themes for analysis, as detailed in Table 1. To construct the analytical framework, we mapped the relevant interview questions and responses across the overarching themes and sub-themes, using MS Excel. Then the team familiarized with the data and created additional codes and sub-codes for the analytical framework. The process was iterative and discussed among three researchers (MM, TM, KS) who conducted the coding and analysis. Once the analytical framework was defined, the researchers imported it into Dedoose software [33] and coded all 82 interview transcripts afresh. Additional codes were added as needed, and memos were used to reflect on themes and comment on the suitability of the analytical framework. Coded data was then exported back and charted in MS Excel, in a matrix summarizing findings per country per theme and sub-theme, as well as the interpretations aligned to the political prioritization framework. Thus, our approach was aligned with framework analysis methods [34] through the key processes of data familiarization, mapping, coding and analytical framework development, indexing, charting and interpretation.

To ensure credibility and trustworthiness, the study relied on an established conceptual framework to analyse the data. Data was sourced from four original case studies that used similar tools and frameworks, through collective effort of the researchers that conducted the original studies. The comparative analysis methodology was developed in consultation with the researchers who conducted the original case studies in each setting, who also provided inputs on the findings and interpretations of the study.

**Table 2. Numbers and types of participants across case study countries.**

| Country | Number of key informants | Stakeholder Groups |
|---|---|---|
| Nigeria (Yobe State) | 19 | Federal and State Ministries of Health (n = 7), multilateral, NGOs, local/civil society organizations (n = 7), professional associations, public and private health providers (n = 5). 12 male and 7 female participants. |
| Somalia (national) | 20 | Federal Ministry of Health (n = 3), donors (n = 2), multilateral and international non-governmental organizations (n = 7), local non-governmental organizations and professional associations (n = 3), and public and private healthcare providers (n = 5). 10 male and 10 female participants. |
| Democratic Republic of Congo (national and North/South Kivu) | 24 | The MoH, the Ministry of Planning, the Universal Health Coverage Fund and the Ministry of Budget (n = 10), bilateral and multilateral organizations (n = 4), researchers (n = 2), North and South Kivu civil society representatives (n = 4), and healthcare providers North and South-Kivu health districts (n = 4). 5 female and 19 male participants. |
| South Sudan (national) | 20 | Government policy makers (n = 5), public and private health providers (n = 4), development institutions (n = 4), international NGOs (n = 4), national NGOs/ civil society organizations (n = 2), and professional associations (n = 1). 13 male and 7 female participants. |

## Reflexivity

In this comparative analysis, the researchers are taking an approach that uses frameworks and common concepts to understand nuanced contexts. However, the objective is not to establish definitively whether or not whether or not MNH is prioritized in each setting, instead describing the contextual nuances and drivers, presenting political prioritization as multifaceted and complex. The majority of researchers involved in the study are based in the countries of interest. Where findings contradict what the researchers may think they know about the context, we emphasized the 'veto power of the sources' [35] and thus relied more on the interviewed stakeholders' own perceptions and experiences.

## Findings

The analysis revealed four key factors influencing political prioritization of MNH in the four settings. Under issue characteristics (i), our study shows how research and evidence shapes MNH political prioritization. In addition, respondent perspectives of how to enhance effectiveness of interventions are discussed. Under political context (ii), we compare policy windows for MNH political prioritization across settings, highlighting the role of crises and political shifts. We also discuss governance structures related primarily to MNH financing and distribution of resources. Actor power (iii) encompasses leadership, policy community cohesion, guiding institutions and civil society mobilization. In the table S2 Table we summarize these findings across themes and sub-themes and per country in more detail. An abridged version is also included in this results section (Table 3).

I. Issue Characteristics

### Role *of* research and evidence *on* MNH political prioritization

Research and evidence contribute credible indicators that can help shape stakeholders' understanding of the features of the problem – that is high maternal and newborn morbidity and mortality. In the context of fragility and humanitarian

**Table 3. Abridged themes and sub-themes emerging from analysis of MNH political prioritization factors across the Shiffman and Smith (2007) framework.**

| Theme/Category | Political priority factor | Sub-themes |
|---|---|---|
| Political context | Policy windows | Favourable events, influence of crises, and role of politics |
| | Governance structure | Financing mechanisms |
| | | |
| Ideas | Internal Frame | Perspectives about MNH as a priority and changing societal attitudes |
| | | |
| Actor Power | Civil society mobilization | Civil society mobilization |
| | Guiding institutions | Accountability, organizational capacities, and international actor influence |
| | Leadership | Limitations on actor authority, influence of local actors, selection of leaders, and the role of women |
| | Policy community cohesion | Research and evidence networks, NGO actor power and relationships, and the quality of actor collaborations |
| Issue Characteristics | Credible indicators | The need for and availability of evidence and evidence based decision making |
| | Effective interventions | conditions for effectiveness of interventions, feedback mechanisms available, provider capacity and quality of care. |

crises, evidence from the four settings show that MNH is not as easy to promote as other health and social issues such as security, disease outbreaks and environmental disasters. However, stakeholders acknowledge the severity of maternal and newborn morbidity and mortality. Evidence plays a key role in decision making, with policymakers and implementers relying on population surveys, independent assessments, and routine health information systems.

*For years we've been talking a lot about evidence-based decisions, but I think there's now a nuance that's been introduced over the years, which is that these are evidence-informed decisions…there are other criteria, other considerations that can also be taken into account when making a decision, and …decision-making is therefore practically a contest between several sets of arguments…Those who make decisions also have other agendas that they absolutely must take into account, otherwise things won't work out. (Researcher, DRC)*

There is some level of evidence use by political figures at national and subnational levels to design policies, ratify international commitments and funnel resources to MNH. Besides research and evidence, there are other factors that drive political prioritization by governments and NGOs. Key informants stated that governments are also driven by issues that cause public outcry while international NGOs focus on issues that are likely to demonstrate big impact due to large numbers of potential beneficiaries.

According to key informants, explanatory research is needed around effective strategies for MNH intervention in the context of re-occurring crises (DRC and Somalia). They also emphasized the importance of more evidence on the community level of care, uncovering barriers to demand and the impact of community level service provision on outcomes (Nigeria, South Sudan). Key informants also identified a need to coordinate evidence collection across different actors, digitalizing and producing timely population surveys. In addition, a gap in country ownership of the MNH research agenda as well as knowledge translation for evidence-informed decision making was identified. Across settings, there was a need to strengthen local research centres, research capacity and dissemination platforms.

**Effective, system-wide interventions**

There was consensus among key informants about cost-effective interventions needed to address mortality and morbidity. These encompassed action on the health system building blocks as well as structural and social determinants of health. System wide approaches were preferred over vertical interventions that could produce negative unintended consequences. For instance, while free maternal care services may be expected to increase uptake, key informants from Somalia stated perceptions of poor quality of freely available services created a preference for private providers. Additionally, key informants from DRC and Nigeria stated that the effectiveness of free healthcare was currently not maximized, as the system building blocks were still inadequate. This situation created a burden of demand for maternal and newborn healthcare without the necessary supplies and inputs to meet it.

Across settings, the critical role of local (including religious) leaders in encouraging or even discouraging uptake of MNH services was emphasized. In Nigeria, for example, local and religious leaders were relied upon by communities to ensure health facility accountability to quality of care, serving as conduits for feedback on behalf of patients and ensuring remedial action. This was particularly important in the contexts where women and girls did not feel empowered to complain directly to providers, even if mechanisms to do so were available.

*"The issue of relatives is our major challenge. They complain about not being allowed time with their relatives. So now, we do allow them because a report came from the king's palace about how women are treated badly. He eventually came around to see for himself and asked all the patients whether they had a relative with them. That one got solved."* (Healthcare worker, Nigeria)

*"We used to have a local meeting with the village heads here to address some of these issues. So… that's the mechanism that we used to get feedback…they used to come and discuss with the chief nursing officer."* (Policymaker 2, Nigeria)

Key informants also emphasized the need to improve the working conditions and empower health workers, particularly midwives, to dedicate focused attention to MNH service delivery. The profession was perceived to be gaining more traction in DRC, while in Nigeria and Somalia, inequities in midwife distribution and retention in hard to reach or rural areas needed to be address.

II. Political Context

**Policy windows**

Key informants discussed events such as onset of crises and political developments that presented as policy windows, creating conditions for prioritizing MNH nationally and sub-nationally. In Yobe State, Nigeria the Boko Haram insurgency prompted the creation of policies to ensure continuity of care through training of midwives originating from the affected localities. This was complemented by the development of politically supported legal frameworks that set the foundations for system strengthening efforts, including the establishment of state ambulance services and tertiary level maternal, newborn, and child healthcare units. Furthermore, political actors declared a state of emergency for health in 2017/18, which led to more investments in health. These efforts have since been complemented and sustained with donor funding, while systematic performance monitoring continues to incentivize political will and government investment in health. Another key policy window was observed in DRC (see Box 1), where a political manifesto by the president in 2019 established frameworks and institutions towards universal health coverage.

Besides political events, health emergencies and other crises have also served as policy windows to generate more investments in MNH. Health emergencies have a reciprocal relationship with political stability and can compel political

actors to act and address poor health outcomes to maintain power or influence system change and effectiveness [36,37]. In Somalia, funding for perennial droughts led to the construction of facilities or expansion of existing ones to accommodate the increased burden of disease and influx of displaced people, with indirect benefits to MNH. Emergencies also necessitated collaboration with the private sector to ensure improved access to care, enhancing reporting of indicators and subsidization of services. In DRC, a similar trend was also observed where emergency funds for Ebola and COVID19 were employed to make infrastructural improvements, leading to reproductive health self-care scale up and risk contingency planning in the health system. Despite the potential of crises as a policy window for improving MNH services, their effect is still largely negative, through diminished capacity for service provision and diversion of resources to other pressing health and non-health emergencies.

---

### Box 1: Role of political champions

In DRC, the President's wish for free maternal care "obliged" technocrats to set up the system around it. He expressed the need for universal health coverage as part of his development plan in 2019, and instituted a special advisor as soon as he was sworn in. This led to joint commissions between the presidency and the MOH for strategic planning, and international support by the head of WHO, who personally met the president and enjoined his organization to supporting DRC in the plan. The MOH also cascaded information down to provinces in the policy planning process, and soon more partners were involved, a process which required harmonization. At the same time, fuelled by the power of the presidency and the WHO, an international UN conference was held in the country in 2020 on UHC commitment. Soon the plans for DRC were interrogated by those that felt their role in UHC implementation was not adequately ascribed, especially financial and technical partners. It took political leadership to emphasize UHC as multisectoral and to insist on the harmonization approach that reflected the president's vision. By 2021 legal ordinances, the strategic plan, and relevant committees had been formed. This kind of high-level leadership set ups the structure in which system transformation can happen that benefits MNH, but one of the respondents cited a common challenge with leadership and governance and policy uptake at subnational levels. Leadership can influence how well anchored projects, even those funded by international aid, are to the government structures if it's not "distracted". Conversely, political leadership can impede achievement of some other objectives including proper partner coordination, system wide approaches, and management of resources among others.

---

Besides crises, there were contrasting findings for political transitions as potential policy windows for MNH political prioritization across the four settings. While the Head of State of the DRC championed free MNH care under universal health coverage as part of his political manifesto from 2019, respondents from Somalia stated that MNH and reproductive health issues did not feature heavily in political discourse during election cycles. Transitions between political regimes may present both positive and negative effects. Sometimes a new leader can bring fresh championship of a health issue, while transitions and reshuffles can also mean a clash between political interests of incoming actors and existing political and technical plans. Key informants from Nigeria stated the need to protect, by statutory measures, established MNH programs to ensure their continuity during transitions. Thus, political transitions in fragile settings can only represent a window for positive change in MNH if there is sustained political championing of the issue, frameworks to enable implementation and ensure continuity, and alignment with national technical plans.

**Governance structures**

In this study, we explored the degree to which established norms, institutions and structures provide effective platforms for collective action on MNH issues. The degree to which collective action can be leveraged for MNH political prioritization

depends on the actor relationships, their relative power and collaborative decision-making. Across all four settings, international aid organizations played a critical role in funding the health sector, thereby influencing its governance. However, national leadership, budget allocation and policy implementation on health was deemed inadequate by respondents. To distribute resources, international actors tend to organize according to "territories" – such as humanitarian zones in Somalia and state demarcations in South Sudan. This compounds a challenge for national authorities to coordinate distribution of services efficiently, including financial pooling and harmonization of programs. To complicate matters further, there is a continuous influx of new actors. For example, in Somalia, new bilateral funders from Arab states added another layer of complexity to the governance and financing landscape.

Governments in fragile settings tend to play a complementary role to international actors in health, i.e., by co-funding programs (Nigeria) and developing multisectoral health support instruments (such as the Health Promotion Fund and Health Solidarity Fund in DRC). According to donor key informants, development funding is meant to be catalytic, aimed at increasing government spending and ownership. However, in practice, it often leads to persistent domestic financial deprioritization of health programs in the long term.

> *"where the priorities are linked with international targets, the funding becomes more readily available. Take for example, the polio eradication is a global target so resources are available… Also for immunization…at the moment, there is very little interest in investing in health* [in general]*, because the government sees other sectors are more priority than health, at least health is taken care of by the donor community so then let us concentrate elsewhere and then later on we come back to it. That later on is never coming." (Policymaker, South Sudan)*

Across settings in this study, MNH is framed largely as a cross-cutting systems strengthening and development issue. As the study settings reflect the development and humanitarian nexus, key informants did not adequately discuss how the MNH agenda can be locally owned and sustained in that context. This is key in the dynamic settings of re-emerging and new crises.

III.  Actor Power

**Leadership**

Government health leadership consists of politically appointed directorship positions and regularly employed technical staff. Technical personnel are regarded as essential to the continuity of health programs during political transitions. However, bureaucratic politics [38] can interfere with day-to-day functions, influencing resource distribution and appointments of technical staff. Respondents in DRC and South Sudan elaborated on how appointed positions can lead to poor leadership and governance when underqualified persons are elected for political expediency. As a respondent from DRC stated, *"unfortunately political interference, as I said earlier, in a country where governance is a problem, you have middle-level managers who have been chosen fresh from university"* (Healthcare worker, DRC). Another respondent from South Sudan echoed:

> *"Let me say in our area, it is a little bit complicated; the way the decision makers are nominated or selected for this job in my place is through… government appointment… that one works well sometimes but sometimes it doesn't work well. Appointing someone can allow anyone to come and give services or to come and give directives to the health workforce but he has no knowledge. But it depends on the appointment authority. If they follow guidelines like for the health department [to] be led by a medical doctor… If we follow these steps, then they will come with their knowledge and they will behave well to direct the health staff in good ways and they will also be respected because they know what they are doing, and they can plan well. So, this is our main challenge." (Healthcare worker, South Sudan).*

In Yobe State, Nigeria, religion also played a role in the appointment of health sector leaders, with Christian civil society actors feeling inadequately represented in decision-making tables in a predominantly Islamic state. Religious representation influenced how issues are framed, and the solutions proposed including service planning, personnel hiring and distribution, and community engagement modalities. While there might synergies between Islam and Christianity on the importance of MNH, health service expectations and preferences may differ based on different restrictions around childbirth and women's participation in public life [39,40]. Thus if representation is unequal in decision making tables, it means preferences of Christian patients may not be represented.

One of the crucial elements of leadership is the capability to unite policy communities and navigate power dynamics among actors to achieve health goals. In fragile settings, national and sub-national governments have to negotiate program priorities with international partners, who are perceived to have more power and influence. A civil society respondent from DRC echoes sentiments from other key informants across settings:

*"The NGO comes with funds, the hand that gives is always superior to the hand that receives, and there are things that at government level we have to say to NGOs: "We want it this way", but that people don't say. So you'd think that the NGO is superior to the government, and that also creates imbalances. They'll do things that aren't allowed and that the government sometimes tolerates". (civil society, DRC)*

Our findings suggest that in a dynamic humanitarian-development context, international and local partner organizations are having to more meaningfully involve governments in their planning to ensure efficiency and sustainability (Somalia). However, as observed by key informants in DRC, some donors and international partners still bypass government structures and local organizations and directly reach beneficiaries. On the other hand, governments have the power to use national legislative frameworks and sociocultural norms to restrict partner organization mandates, such as on abortion.

Across the countries, key informants also talked about how beneficiaries, religious and community leaders also exert influence by accepting or refusing services, shaping demand and influencing quality of care. There were positive and negative examples of this bottom-up influence, such as the community leaders in Yobe State conducting facility oversights to ensure quality of care. In DRC however, local leaders were blocking regulation or monitoring of private facilities, leading to inadequate quality assurance. Local organizations are by and large not seen as empowered, lacking resources and capacities for high-level engagements.

The role of women in health sector leadership was influenced by national and international efforts in gender equity and mainstreaming, as well as prevailing gender and social norms. Female leadership was more common in international partner organizations and national policy institutions. Donor funding requirements, backed by national laws and regulations, ensured the implementation of gender equity frameworks. National laws and regulations are related to quotas for women's political representation in legislative and other decision-making spaces. Despite these representations, key informants noted that there were some challenges to be addressed. For instance, stakeholders in South Sudan and Somalia felt that women were not able to use their voices effectively in decision making platforms, influenced by the norms of male-dominated cultures. This shows that involving women in these spaces may be a political choice against predominant norms as well as an exercise of their own agency. Key informants also emphasized the importance of skills and competencies beyond gender representation. As a key informant from Somalia stated, *"It is not necessary for women to be present at decision-making tables; rather, they must be well informed and knowledgeable about the situation of MNH in Somalia"* (*Healthcare worker*). However, informants did not reflect on political and sociocultural choices influencing lack of education, skills and competencies among women.

In addition to skills, stakeholders (who in this study were largely men) believed that women should possess the necessary assertiveness or leadership capacity, placing the onus to have their voices heard on them:

*"there is a growing trend for more women getting opportunities. For example, if you take {UN agency}, now most of our decision makers; executive directors, regional directors, county representatives and deputies are almost all women… The problem now is when it comes to civil servants, like the politicians… it is like ticking the box sometimes. You see them there but they don't make decisions. At times it could be even their weakness because if given the opportunity, why not use it?". (NGO, South Sudan)*

*"S,o I think insofar as there are women who raise their voices to speak, they are listened to. But what's less note-worthy is that there are very few women who have the courage or perhaps the skills to make their voices heard at decision-making level. I think that when they are there, we try to take into account their presence and their opinions" (Researcher, DRC).*

Other stakeholders viewed women as having the unique positionality to take the mantel of the MNH agenda and bring different perspectives to decision making tables. Women at high levels of decision making were normally aligned with gender and MNH departments, which could be leveraged to champion and bring nuance to the framing of issues and improvement of MNH programs. At facility level, key informants (DRC and Nigeria) highlighted a lack of female leadership where gender hierarchies were prominent, with nurses who are often female and facility-in-charges who are often male.

While key informants discussed positive progress in gender mainstreaming over the last decade, it does not automatically translate into positive dynamics in communities. A respondent from South Sudan cited that gender-based violence continues to occur even as local initiatives involved more women in the health workforce and local decision-making bodies. Gender and social norms thus continue to be entrenched and act as a barrier to transformation, particularly at household and community levels. A key informant from Somalia elaborates:

*At the leadership level, women do play a role, but they cannot change or do something about traditional biases that bar women to make their own health decisions… the issue of male consent is very tough on mothers' health. I encountered many cases where the husband rejected the consent for surgery to save the life of his wife. If you operate on the mother without the male consent and she dies, you will be taken to court and that is a difficult obstacle"(Private Health Provider, Somalia).*

Lastly, there was a general lack of representation by women-led organizations in decision making across all settings. Stakeholders cited poor organization, capacities and prevailing gender biases as reasons for poor engagement of women-led local organizations. Some approaches highlighted to improve women's participation in leadership were discussed and include Boma Health Facility Boards or Village Health Committees in South Sudan, and Ward Development Committees in Nigeria. However, the integration of these practices across the system and their effectiveness in addressing MNH issues is unclear.

### Guiding institutions

Technical working groups and multi-stakeholder health committees at national and subnational levels are crucial guiding institutions for the health sector and program design in fragile settings. International donors and implementing organizations are central to these, engaging in national policy processes for MNH and reproductive health. A common challenge was the sustainability of the designed programs and their ability to remain in alignment with national priorities. Coordination was reported to be especially poor at subnational level (South Sudan and Nigeria). Some of the coordination mechanisms such as technical committees were also temporary depending on the level of funding or the passing of an emergency. Through these committees, governments attempt to ensure alignment and coordination of partners and resources, although this is not always adequate or consistently done. The ability to effectively coordinate is related to the power held by the accountable actor, in this case governments.

The challenge of governing the health sector and coordinating partners was a common theme across all settings. Besides underfunding, one of the factors that weakened the government capacity was the extraction of skilled personnel from government to partner organizations. This indirectly impacts national MNH political prioritization through a general weaking of capable technical leaders in government. In both South Sudan and DRC, key informants highlighted the turnover of competent individuals from health ministries to partner organizations. As a respondent from South Sudan states,

*"We are currently seriously understaffed. The reason is that the Ministry of Health is challenged to retain these staffs… As a result, we have very few staff that are available…there are some with very good capacity but there are some where we still need to build the capacity…we are constantly losing those who have adequate capacity because the moment we build their capacity they become more marketable. The partners are the ones getting the funding so they keep depleting the Ministry of staff…That is why we complain of the implementation arrangements when we are using the donor funding mechanism that uses third party as implementers. They invest now in building their capacity rather than strengthening the national health system." (Policymaker, South Sudan)*

Key informants from South Sudan also described initiatives by some international partners to invest directly in government capacity and ownership by moving away from the more common third party implementor models.

Across settings, reporting systems were weak and poorly integrated. Governments and partner organizations sometimes collaborated to ensure accountability for MNH outcomes. Accountability mechanisms such as joint supervisions were challenged by lack of sustained implementation or dedicated resources to sustain them. When adequately functional, localized reporting systems are then linked to global reporting mechanisms in donor and multilateral organizations. Respondents stated that other influential actors such as the private sector were not adequately integrated into these mechanisms. As it is, reporting systems remain fragmented – with government reporting systems weaker given their low resources and investments.

## Policy community cohesion

Across settings, key informants discussed challenges in the cohesiveness of policy communities, affecting their inclusiveness and ability to maximize resources, garner political support and ensure broad coverage of interventions for MNH. Respondents from South Sudan and Somalia described challenges in programming including duplication of effort (where multiple organizations implement in the same locations) and fragmented decision-making. Conflict disrupted the working of policy communities and civil society actors were poorly involved in decision making, leading to poor representation of grassroots level understanding of issues.

While governments were recognized as key to MNH policy development and implementation, there was a perception that their role was mainly to provide political support for issues that are lobbied to them by donors and partners. Governments lacked the resources to fully own the policy agenda. Despite the acknowledgement of the differential power dynamics with international partners, the critical role of government in the cohesion of the policy community was emphasized by key informants. Governments facilitated entry into communities and created an enabling policy environment for implementation of MNH programs. Conversely, governments could also negatively impact cohesion by bestowing political favour differentially across partner organizations and perpetuating competition:

*"On our side the donor agencies, multilaterals, nongovernmental organizations… it's time we start thinking about one voice. One voice in advocacy, one voice in communication… they seem to be talking to each other, you know they can only get territorial if they're encouraged to. Most especially if the past government takes preference of one over the other" (NGO, Nigeria)*

Policy community cohesion was also negatively impacted when international actors bypassed existing structures, government or local organizations, creating parallel systems for implementation of programs.

*"… we sometimes find other mechanisms to reach these beneficiaries… if we think we need to work with such and such a structure to reach the population, if the structure doesn't manage to do so, we try to bypass it to reach the beneficiary. For us, it's the beneficiary who's interesting. It's not the structure that's there…And bypassing it means finding another structure, or finding another way to reach the same populations." (Multilateral organization, DRC)*

Cohesion was also be threatened by the push for international policies, which according to respondents were sometimes accepted by governments without sufficient scrutiny (DRC). The perception among some respondents is that governments needed to be "pushed" to co-sign policies and programs, which partners then implement. There was not much discussion of government taking ownership of the MNH agenda and 'pulling' partners to its cause. A key informant from South Sudan highlighted how even the research agenda was advanced by partners, when it should be the Ministry of Health leading efforts; since they possessed the authority to develop and implement the subsequent policies.

**Civil society mobilization**

Across all countries, civil society groups played a key role in community entry and sensitization, advocacy and other program support. However, civil society actors felt excluded from high level decision-making and inadequately funded to enable capacity building and more meaningful engagement. In Somalia, for example, there were mixed responses regarding how much civil society was involved, with policy makers claiming they were involved in decision making while some stakeholders claimed they were not. Key informants discussed poor linkages of civil society actors to relevant networks and poor capacity to engage where other partners may have more power, financial resources or technical knowledge. The question of capacity was also discussed by informants from DRC, while also recognizing their importance in advancing community level maternal death surveillance and sensitization about services.

IV. Ideas

Overall, key informants perceived that MNH was a priority in their settings in terms of the presence of dedicated partners, government commitments and allocation of resources to programs. However, challenges remain. For instance, free maternity services under universal health coverage were not fully or consistently implemented in neither DRC nor Yobe State, Nigeria. Political prioritization was also seen to fluctuate with waning funds for the MNH agenda nationally and internationally.

At the community level, key informants stated that MNH was a priority in terms of the value placed on mothers, children and family in local cultures. However, despite those perceptions, they also acknowledged that the achievement of better health status is complicated by poor health seeking behaviours, unequal gender power relations and poor access to healthcare services. The sentiment among key informants was that attention is increasing, due to various contextual developments such as community health system strengthening (South Sudan), community midwife training and recruitment and increased financial allocation to health (Nigeria), and better uptake of MNCH services (Somalia).

Changes in cultural attitudes and efforts in gender transformation were also starting to show results; these include the engagement of men in MNH in South Sudan and empowerment of couples and women in the uptake of family planning and facility-based care in DRC. Respondents noted that this increased acceptance was particularly observed in urban areas, where women were also increasingly participating in decision making around MNH policy and programs. Other attitude changes that may positively influence MNH, as observed by key informants, included enhanced political ownership of the health agenda (Nigeria and DRC), which led to more investments, health promotion and contextual adaptations to maximize intervention effectiveness (e.g., the employment of indigenous female midwives).

## Discussion

This study explored key components of political prioritization for MNH across four fragile settings in Africa. The study captured multiple perspectives from different levels of the health system, from healthcare workers in the public and private sector to national decision-makers from governmental and non-governmental institutions. The alignment of perspectives across settings offered generalizations that could be useful for understanding other similar fragile contexts, while nuanced differences where highlighted where relevant.

Across settings, the severity of maternal and newborn morbidity and mortality was recognized as a problem. Key informants, however, highlighted that MNH was a challenge among many others, competing with health and other priorities for the attention of decisionmakers, as is common in fragile settings [41]. The investments that need to be made in these settings are thus all-encompassing, geared at ensuring overall health systems resilience [42]. The problem of competing priorities manifests in the lack of domestic financial allocation for health in all the settings, particularly in Somalia and South Sudan where contributions were much lower than the 15% recommended in the Abuja declaration [43]. In these settings, external funders bolstered the health sector.

In our study settings, investments in MNH as a priority were generally indirect, through general system strengthening approaches. One example is the piloting of free MNH services in DRC as part of the country's focus on universal health coverage. Aid funds are often directed to infrastructure, equipment and supplies that indirectly benefit MNH outcomes. However, the confluence of competing health priorities in fragile settings does not necessarily have to impede MNH investment: In Nigeria, compounding crises and declaration of health state of emergency resulted in infrastructural and human resource investments specifically to ensure MNH continuity of care. A level of vertical investment as seen in Nigeria enables scaling of interventions to reduce the burden of morbidity and mortality [44], and may be beneficial in weak health systems with inadequate human resources and poor reach to vulnerable populations [45]. For long term sustainability however, stakeholders in fragile settings recognize the need to balance with system wide investments.

Although complex and recurring crisis marked the fragility across settings in this study, they also created policy windows or opportunities for strengthening and transforming health systems. This includes the previously mentioned Yobe State health state of emergency in 2017/18 [46] as well as the Ebola epidemic that struck a province in western DRC in 2018, both of which catalysed free (limited) healthcare provision. An evaluation of the DRC policy saw some improvement in MNH indicators, albeit short lived during the project period (a negative consequence of building health systems using surge funding) [47]. The international momentum around universal health coverage (UHC) also created a window of opportunity for better political prioritization of MNH [48]. This momentum coincided with the election of DRC president in 2019, who committed to free MNH services under UHC. Thus, while crises also present setbacks, the related surges in funding and election of political champions create opportunities to transform the health system and MNH services.

Key informants from other countries in this study did not discuss UHC, but a WHO report demonstrated a need for it across the settings, highlighting catastrophic spending on health and poor coverage of services [49]. The countries in this study have had political commitments for UHC in the past, as well as roadmaps and enabling legislative and policy frameworks for implementation [50–52]. For low-income countries also facing fragility, guidelines show that UHC can be achievable with measures that include an increase of 1% of GDP in domestic allocation to health, focusing on primary care and health systems strengthening, and reduction or waiver of user fees. While global health agendas like UHC can create policy windows for better political prioritization of MNH, localization is still needed to sustain outcomes [53].

In this study and others, community-based approaches demonstrated potential to enhance access to MNH care, promote health and improve health workforce availability [9]. However, community-based intervention needs to be coupled with a commitment to build local capacities to respond to health needs, which is a challenging barrier in fragile settings [54]. This study found that sometimes actors bypass local organizations, which may increase efficiencies in the short term but entrenches inequities in the long term. A study from South Sudan and Haiti demonstrated how community health

programmes can influence local ownership through bottom-up approaches, enhancing trust and legitimacy, and influencing sustainability [55]. In those settings, trust-building activities with local communities (including sensitivity to the ethnic divisions responsible for conflict), indirectly strengthened social cohesion through responsive programming and operational mechanisms.

Unequal power relations and the political nature of health leadership explained challenges in health system governance, resource distribution, and inclusivity in decision making in the study settings. International actors contributed more resources to health than local governments, with negative and positive influences observed. Fragmented projects by international actors undermined potential for any sustained health impact, exacerbated by short term humanitarian funding and weak health governance systems. Conversely, positive contributions to gender equity and health policy development were observed. Coordination mechanisms that could address this fragmentation continue to be marred by unequal power dynamics, weakened government capacity, and inadequate accountability efforts. Our findings thus suggest that the effectiveness of guiding institutions and coordinating mechanisms can be ensured by minimizing ways of working that entrench the dominance of powerful players. More research is needed evaluating the effectiveness of new models of engagement proposed in this study and others including hybrid models [56], nexus frameworks [57], and localization [57]. New approaches such as fragile-to-fragile cooperation can be evaluated for their contribution to high level joint political decision-making, flexible processes and multidirectional flows of funding and resources [58].

Lastly, our findings described the role of women, civil society groups and community/religious leaders in influencing MNH issue framing, and service demand and utilization. Key informants suggested that for women in high level decision making to be more effective, they must be empowered with knowledge, capacity and leadership skills. While this also applies to men in high level decision making, key informants also acknowledged that there were persistent sociocultural norms that impede women's leadership across levels of the health system. While women's participation can be ensured by gender equity and mainstreaming efforts, these programmatic approaches may also miss the opportunity to transform broader societal and health system structures [59].

Besides women, civil society groups across all settings were inadequately involved in decision making, even while their role in community entry, advocacy and program support was recognized. Civil society engagement has the potential to improve health sector performance if adequately integrated in long term development plans by international and national actors [41,54]. More research is needed on modalities for effective engagement of civil society in MNH and related agendas. And lastly, respondents emphasized the pivotal role of community, religious and other traditional leaders in the uptake and quality of MNH services. While similar findings were observed in other low-income contexts, evidence also shows that in hierarchical/patriarchal cultures the role of local leaders may also be coercive; and thus bottom-up, empowerment-based approaches need to be explored [60].

## Limitations

This study reflects the views of current stakeholders in MNH policymaking and implementation in the respective settings and may not comprehensively cover all issues related to political prioritization. Because of the secondary data analysis, we were unable to capture perspectives related to the external framing of MNH, as it was not explored in the original studies. Components of the Shiffman and Smith framework were also discussed according to the available data; thus, future research can delve deeper into under-discussed components such as civil society mobilization and the role of ideas in MNH political prioritization. In additional only state level data was available for Nigeria, reflecting the focus in the primary research on the conflict affected parts of the country. Since the settings of this study are dynamic (such as the ongoing conflict in Eastern DRC at the time of writing in early 2025) changes in contextual factors may influence some of the findings of the study.

## Conclusion

This study presented stakeholder perspectives of MNH political prioritization across four fragile settings in South Sudan, Somalia, Nigeria and DRC. MNH was recognized as an important issue but competing with other priorities within the context of complex crises and weak health systems. Respondents identified policy windows created by political crises for transforming health systems and improving MNH outcomes. Politics influenced leadership and governance capacity, resource distribution, coordination of health sector partners as well as continued support for the MNH agenda. Governments and international actors negotiated power, with the latter contributing technical and financial resources, and the former influencing policy, implementation and coordination. Several factors challenged the cohesion of actors, including political influence, unequal resources, and the lack of inclusivity in decision-making. Overall attention is increasing towards MNH, especially in community interventions, gender transformative approaches and political ownership. However, domestic financing for MNH and broader health system strengthening through universal health coverage is still needed, to ensure sustainable resources and impact.

## Supporting information

**S1 Table. Comparison of country cases across political and socioeconomic dimensions.**
(XLSX)

**S2 Table. Summary of key sub-themes emerging from analysis of MNH political prioritization factors across categories of the Shiffman and Smith (2007) framework.**
(XLSX)

## Acknowledgments

We thank the following EQUAL consortium researchers for their contribution to this work: Gishlain Bisimwa, Pacifique Lyabayungu, Emilia Iwu, Rejoice Abimiku, Charity Maina, Mohamed Jimale, Asia Mohamud, Hawa Abdullahi.

## Author contributions

**Conceptualization:** Mamothena Carol Mothupi, Teresia Macharia, Maryan Abdulkadir Ahmed, Abdirisak Dalmar, Rifkatu Aimu Sunday, Paul Spiegel.

**Data curation:** Mamothena Carol Mothupi, Teresia Macharia, Katja Starc Card.

**Formal analysis:** Mamothena Carol Mothupi, Teresia Macharia, Katja Starc Card.

**Funding acquisition:** Naoko Kozuki, Paul Spiegel.

**Investigation:** Mamothena Carol Mothupi, Teresia Macharia, Katja Starc Card.

**Methodology:** Mamothena Carol Mothupi, Teresia Macharia, Rosine Bigirinama, Maryan Abdulkadir Ahmed, Rifkatu Aimu Sunday, Sussan Israel-Isah, Kon Paul Alier, Naoko Kozuki, Paul Spiegel.

**Project administration:** Mamothena Carol Mothupi, Teresia Macharia.

**Resources:** Abdirisak Dalmar, Naoko Kozuki, Paul Spiegel.

**Software:** Mamothena Carol Mothupi, Teresia Macharia, Katja Starc Card.

**Supervision:** Mamothena Carol Mothupi, Abdirisak Dalmar, Naoko Kozuki, Paul Spiegel.

**Validation:** Mamothena Carol Mothupi, Katja Starc Card, Rosine Bigirinama, Alicia Adler, Rifkatu Aimu Sunday, Sussan Israel-Isah, Kon Paul Alier, Paul Spiegel.

**Visualization:** Mamothena Carol Mothupi, Rosine Bigirinama, Alicia Adler.

**Writing – original draft:** Mamothena Carol Mothupi.

**Writing – review & editing:** Mamothena Carol Mothupi, Teresia Macharia, Katja Starc Card, Rosine Bigirinama, Alicia Adler, Maryan Abdulkadir Ahmed, Abdirisak Dalmar, Rifkatu Aimu Sunday, Sussan Israel-Isah, Kon Paul Alier, Naoko Kozuki, Paul Spiegel.

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
