## [Decision Letter · Decision Letter 0]

14 Jul 2025

PGPH-D-25-00559

“Crises are a perpetual restart” – a comparative analysis of maternal and newborn health political prioritization across four fragile and conflict affected settings.

Dear Dr. Mothupi,

Thank you for submitting your manuscript to PLOS Global Public Health. We appreciate your patience during this process. After careful consideration, we feel that it has merit but does not fully meet PLOS Global Public Health’s publication criteria as it currently stands. Therefore, we invite you to submit a revised version of the manuscript that addresses the points raised during the review process.

We look forward to receiving your revised manuscript.

Kind regards,

Veena Sriram

Academic Editor

Journal Requirements:

1. We have noticed that you have a list of Supporting Information legends in your manuscript. However, there are no corresponding files uploaded to the submission. Please upload them as separate files with the item type 'Supporting Information'.

2. In the online submission form, you indicated that “Data can be accessed from the manuscript authors.”.

3. Uploaded as supplementary information.

Additional Editor Comments (if provided):

Reviewers' comments:

Reviewer's Responses to Questions

**Comments to the Author**

1. Does this manuscript meet PLOS Global Public Health’s publication criteria?

Reviewer #1: Yes

Reviewer #2: Yes

2. Has the statistical analysis been performed appropriately and rigorously?

Reviewer #1: N/A

Reviewer #2: N/A

3. Have the authors made all data underlying the findings in their manuscript fully available (please refer to the Data Availability Statement at the start of the manuscript PDF file)?

Reviewer #1: No

Reviewer #2: Yes

4. Is the manuscript presented in an intelligible fashion and written in standard English?

Reviewer #1: Yes

Reviewer #2: Yes

Reviewer #1: Thank you for the opportunity to review this insightful paper on the political prioritization of MNH in fragile and conflict affected settings. It is powerful, valuable and timely research.

Overall, there is substantial relevant data to show the various factors shaping MNH prioritization in these settings. In the Findings, there are some really strong sections presenting clear arguments that respond to the research question, such as the “effective, system-wide interventions” section. However, there are other times when there are several different points being made at once, which makes it difficult to follow the thread of the argument, or understand which are the most salient issues regarding prioritization. Some paragraphs start with a focus on one issue, then jump to another – e.g. the paragraph starting at Line 286, starting with discussion of political leadership and then shifting to environmental/health crises. There are a few other places where there are a few connected ideas, but the link is not made explicitly or the overall point fleshed out, such as with paragraphs starting Line 226 and Line 368 (for instance with the Line 368 paragraph, how does religious representation influenced issue framing and what is its impact on prioritization?). Perhaps some of these are not such significant factors in political prioritization – it would be interesting to know what the data suggests is really key and has consensus within/across informants, versus those with less explanatory power. Finally for the Results, it could be made clearer at times how the different points link to political prioritization specifically, rather than governance generally.

In the Introduction, I appreciate the comprehensive context provided, but it could at times be focused more tightly on MNH policy, its impact and why political prioritization is important to understand/ analyze (especially paragraphs 2 & 3).

In the Methodology section – why are some data national and some state level? What impact does this have on findings? I appreciate the explanation of how the framework is being applied to the case, and am curious whether the findings diverged from the framework in any way (other than not including external framing for practical reasons)? So, for example, if any key categories emerged that could not fit into the framework.

The paper explicitly states the focus is on political prioritization rather than prioritization in general (Line 108), but then at the beginning of the Findings it describes “influencing prioritization of MNH in the four settings” (Line 202). It would be helpful to be really clear about the distinction between the two and which is consistently the focus of the paper throughout.

A couple of more minor points:

Some of the quotes are very long, and could potentially be cut down to make the same point and ensure the reader is clear about the purpose of the quote.

Line 120 – last box of table, incomplete sentence? “May address”…MNH?

Line 281 – MNH or MNCH? If MNCH spell out as don’t think you mentioned before.

Line 349 – “regarded AS essential”

Line 350 – what kind of politics? There are a few times politics is mentioned, but is a very broad term, would be helpful to have more specificity.

Line 459 – “one of THE factors”

Reviewer #2: Thank you for this important work. More attention is needed to healthcare delivery, policy and implementation - and this paper really helps to understand the context based solutions better.

My comments - all minor - were to improve the strength of presentation and make the paper more methodologically stronger just by explaining a bit more.

Please find the comments below:

Overall:

Based on recent critiques from scholars in the African region, I encourage authors to move away from using “Sub-Saharan Africa” given its colonial roots and non-evidence informed terminology. Since the authors have used a mix of “Africa,” “sub-Saharan Africa,” and “African region” – I think it will be easier to remove sub-Saharan Africa without compromising the clarity or quality of the paper.

Please consider revising the term after considering these sources:

https://www.devex.com/news/opinion-why-we-ve-stopped-using-the-term-sub-saharan-africa-105692 (same piece without a paywall: https://www.firelightfoundation.org/blog/why-weve-stopped-using-the-term-sub-saharan-africa)

https://www.africarebirth.com/the-racist-connotations-of-the-term-sub-saharan-africa/

Methodology:

It would be good to say that qualitative analysis was “deductive analysis” since the authors used a framework and established themes to group data.

https://researchmethodscommunity.sagepub.com/blog/qualitative-research-design-and-data-analysis-deductive-and-inductive-approaches

Results:

Since the results section is quite long, would be good to give an overview at the top outlining the themes and what are within. For example, leadership theme under Actor power has multiple themes including gender equity. Highlighting these themes, subthemes at the beginning maybe in a summary table perhaps will be helpful to the reader – also to use this paper for policymaking approaches.

Alternatively, a prose outline of the results can be provided at the beginning of the results section, and a summary table can be provided at the end of results. Summary table can be easily drafted using main sections and subthemes.

I recognize that there is a detailed supplementary table (S2 Table 4), I wonder if this can be repurposed and included in the manuscript. I think it will greatly improve the presentation of results.

Line 462 – might be good to also include a quote from DRC if one is readily available, since DRC is mentioned. It might be good cut down the South Sudan quote a bit, if possible.

This is not a must – but across the results section, when possible, I would cut down on long quotes – without compromising the quality of results. I think some repetition could be reduced. I leave it to the discretion of the authors and their stylistic approach of how they envision to present the results. Generally, readability would improve if you can cut down repetition within quotes, where possible.

**Do you want your identity to be public for this peer review?** For information about this choice, including consent withdrawal, please see our Privacy Policy

Reviewer #1: No

Reviewer #2: No

---

## [Editor Report · Decision Letter 1]

21 Sep 2025

“Crises are a perpetual restart” – a comparative analysis of maternal and newborn health political prioritization across four fragile and conflict affected settings.

PGPH-D-25-00559R1

Dear Dr. Mothupi,

We are pleased to inform you that your manuscript '“Crises are a perpetual restart” – a comparative analysis of maternal and newborn health political prioritization across four fragile and conflict affected settings.' has been provisionally accepted for publication in PLOS Global Public Health.

Best regards,

Veena Sriram

Academic Editor

Thank you for your excellent revision on this article. I am happy to accept this for publication. I did want to note that there are a few typos in a few places (e.g., Coordination was reported to be especially poor at subnational level (South Sudan and Nigeria); (In additional only state level data was available for Nigeria, reflecting the focus in the primary research on the conflict affected parts of the country). Please be sure to review carefully during the proof stage. Congratulations!